# Functional Role of microRNAs in Regulating Cardiomyocyte Death

**DOI:** 10.3390/cells11060983

**Published:** 2022-03-12

**Authors:** Urna Kansakar, Fahimeh Varzideh, Pasquale Mone, Stanislovas S. Jankauskas, Gaetano Santulli

**Affiliations:** 1Department of Medicine (Cardiology), Wilf Family Cardiovascular Research Institute, Einstein Institute for Aging Research, Institute for Neuroimmunology and Inflammation (*INI*), Albert Einstein College of Medicine, New York, NY 10461, USA; urna.kansakar@einsteinmed.edu (U.K.); fahimeh.varzideh@einsteinmed.edu (F.V.); pasquale.mone@einsteinmed.org (P.M.); stanislovas.jankauskas@einsteinmed.edu (S.S.J.); 2Department of Molecular Pharmacology, Einstein-Mount Sinai Diabetes Research Center (*ES-DRC*), Fleischer Institute for Diabetes and Metabolism (*FIDAM*), Albert Einstein College of Medicine, New York, NY 10461, USA

**Keywords:** cell death, cardiomyocytes, ischemia/reperfusion, myocardial infarction, heart failure, autophagy, apoptosis, necrosis, non-coding RNA

## Abstract

microRNAs (miRNA, miRs) play crucial roles in cardiovascular disease regulating numerous processes, including inflammation, cell proliferation, angiogenesis, and cell death. Herein, we present an updated and comprehensive overview of the functional involvement of miRs in the regulation of cardiomyocyte death, a central event in acute myocardial infarction, ischemia/reperfusion, and heart failure. Specifically, in this systematic review we are focusing on necrosis, apoptosis, and autophagy.

## 1. Introduction

MicroRNAs (miRNAs, also known as miRs) are a group of small single-stranded noncoding RNA (ncRNA) with approximately 18–22 nucleotides; miRNAs regulate gene expression by binding to the 3′-untranslated region (3′UTR) of the messenger RNA (mRNA) of a target gene, enhancing its degradation and/or inhibiting protein translation [1,2,3].

Numerous studies have shown that miRNAs play essential roles in cardiovascular diseases regulating a plethora of processes including cell death, cell proliferation, inflammation, and angiogenesis [4]. In this review, we will focus on the involvement of miRNAs in the regulation of cardiomyocyte death.

Cardiomyocyte death is the central event in acute myocardial infarction; additionally, cell death has been shown to contribute significantly to the pathogenesis of ischemia/reperfusion and heart failure [5,6,7].

## 2. miRNAs and Cardiomyocyte Necrosis

Necrosis (from Ancient Greek νέκρωσις, nékrōsis, “death”) is a pathological process characterized by a gain in cell volume, swelling of organelles, lysis of plasma membrane, and leaking of intracellular content [8]. Necroptosis, mitochondrial permeability transition (MPT)-dependent necrosis, pyroptosis, ferroptosis, parthanatos, and NETosis are all different types of programmed necrosis [9,10]. 

Various investigations have shown that necroptosis and MPT-dependent necrosis are involved in the pathogenesis of myocardial infarction, ischemia/reperfusion, and heart failure [11,12]. Necroptosis is initiated by the binding of cytokines including tumor necrosis factor alpha (TNF-α), Fas/CD95, and TRAIL (TNF-related apoptosis-inducing ligand) to death receptors (e.g., TNFR1, TNFR2) and to active receptor interaction protein kinase 1 and 3 (RIP1 and RIP3), causing the formation of the necrosome [13,14].

Pei-feng Li’s research team was the first to report that miR-103/107 induces necrosis through targeting Fas-associated protein with death domain (FADD) in H_2_O_2_-treated H9c2 cardiomyoblasts and, in vivo, in a mouse model of ischemia/reperfusion; FADD inhibits necrosis by blocking the formation of the receptor-interacting serine/threonine protein kinase 1 and 3 (RIP1-RIP3) complex [15]. They also found that the long non-coding RNA H19 (lncRNA-H19), which contains three potential miR-103/107 binding sites, can attenuate cardiac necrosis by inhibiting miR-103/107 [15]. These results support a link among H19, miR-103/107, FADD, and RIPK1/RIPK3 in cardiomyocytes. 

Another long non-coding RNA, namely necrosis-related factor (NRF), was shown to regulate necrosis in H_2_O_2_-treated cardiomyocytes and in mice challenged by ischemia/reperfusion through means of the repression of miR-873; this specific miR reduces necrosis by silencing the translation of RIPK1/RIPK3 [16]. Since NRF is also modulated by p53 [17], NRF overexpression by p53 can markedly aggravate myocardial necrosis [16]. Another study proved that lncRNA E2F1 increases cardiomyocyte necrosis by inhibiting miR-30b and cyclophilin D (CypD); in fact, miR-30b decreases necrosis and infarct size by targeting CypD [18].

miR-874 represents a perfect example of a miR that is mechanistically involved in cardiomyocyte necrosis experimentally induced by H_2_O_2_: its expression increases in H_2_O_2_-treated murine cardiomyocytes and its knockdown reduces necrosis in H_2_O_2_-treated cardiomyocytes and in a murine model of ischemia/reperfusion injury; caspase-8, which negatively affects myocardial necrosis by cleaving and inactivating RIP3 [19], was identified as a downstream target of miR-874 [20]; FOXO3a is known to repress miR-874 expression [21] and its overexpression attenuates cardiac necrosis [20].

In a mouse model of myocardial infarction, miR-325-3p overexpression was shown to suppress the expression of necrosis mediators including RIPK1, RIPK3, and phosphorylated mixed lineage kinase domain-like protein (p-MLKL) [22]. Reduced expression of lactate dehydrogenase (LDH), phosphocreatine kinase (CK), superoxide dismutase (SOD), and malondialdehyde (MDA), and increased left ventricular ejection fraction (LVEF) and left ventricular fractional shortening (LVES) were observed by overexpressing miR-325-3p [22]. An increased expression of miR-155 was detected in cardiomyocyte progenitor cells when stressed with H_2_O_2_ [23]; miR-155 overexpression decreases necrotic cells through inhibiting RIP1 [23,24]. 

The downregulation of Adenine nucleotide translocase 1 (ANT1) is another mechanism underlying cardiomyocyte necrosis [25]; by directly targeting ANT1, miR-2861 was shown to induce necrotic cell death in hypoxia-treated murine cardiomyocytes, as well as in mice challenged by ischemia/reperfusion injury [26]; confirming these findings, knocking miR-2861 down preserved cardiomyocytes from necrosis via inhibition of ANT1 degradation [26].

## 3. miRNAs and Cardiomyocyte Apoptosis

Apoptosis (from Ancient Greek ἀπόπτωσις, apóptōsis, ‘falling off’) is a form of programmed death [27,28] that plays an important role in the pathogenesis of cardiovascular disorders [29,30,31,32]. In conditions such as myocardial infarction, apoptosis occurs in terminally differentiated cardiomyocytes and inhibition of apoptosis has been shown to protect the heart against ischemia/reperfusion injury [30,33].

The apoptosis machinery is activated by intrinsic mitochondrial pathway and extrinsic cell surface death receptor pathway and is characterized by a series of structural and morphological changes including chromatin condensation, DNA degradation, cell shrinkage, and blebbing of plasma membrane [34,35]. In the extrinsic pathway, extracellular ligands, such as tumor necrosis factor alpha (TNF-α) and FasL bind to the death receptor (e.g., Fas, TNFR) to activate intracellular caspases [36]. 

On the other hand, the intrinsic pathway is mainly mediated by mitochondria-associated BCL-2 family proteins, namely BAX and BAK [37,38,39,40,41,42], located in the outer mitochondrial membrane [43]. Intracellular stress signals, such as oxidative stress, hypoxia, or DNA damage induce BAX and BAK to release mitochondrial proteins, cytochrome c (a well-known activator of caspases [7,31]), and DIABLO into cytoplasm [44,45].

In the recent years, we and others have demonstrated that miRNAs have an instrumental role in apoptosis and in the pathogenesis of myocardial disorders [1,46,47,48,49,50,51,52] (Figure 1).

miR-133a is one of the fundamental miRNAs able to enhance the survival of cardiomyocytes in anoxia and hypoxia conditions. Indeed, miR-133a inhibits apoptosis via repressing Transgelin 2 (TAGLN2), chaperonins heat shock protein 60 and 70 (HSP60, HSP70), apoptotic protease activating factor-1 (Apaf-1), caspase-3/8/9, and by indirectly enhancing Bcl-2 expression [53,54,55,56]. Instead, miR-1 is downregulated under hypoxia, and its overexpression elevates cell apoptosis by reducing Bcl-2 [57].

The upregulation of miR-122 has been reported in rat cardiomyocytes after hypoxia-reoxygenation treatment; miR-122 overexpression promotes cell apoptosis via targeting GATA-4 [58]. Liu et al. found in H9c2 cardiomyoblasts that miR-208a triggers apoptosis through silencing activated protein C (APC) in hypoxic conditions; APC reduces apoptosis in hypoxia and knockdown of APC attenuates the inhibitory effects of miR-208a [59]. A recent study revealed that the upregulation of miR-137-3p can aggravate cardiomyocyte apoptosis induced by ischemia/reperfusion through means of the downregulation of the Kruppel-like factor 15 (KLF15). [60]. In vitro, a significant downregulation of miR-7a-5p was detected in H9c2 cardiomyoblasts undergoing hypoxia-reoxygenation; this finding is noteworthy, inasmuch as the overexpression of miR-7a-5p inhibits the expression of cleaved caspase-3 and Bax and promotes the expression of Bcl-2 by targeting voltage-dependent anion channel 1 (VDAC1) [61].

Wu et al. demonstrated that miR-613 attenuates cell apoptosis by targeting Programmed cell death 10 (PDCD10) in cardiomyocytes undergoing ischemia/reperfusion [62]. Hypoxia treatment also augments miR-9 expression levels, and miR-9 enhances cell apoptosis via repressing CDK8; in fact, knockdown of CDK8 reverses the inhibitory effects of miR-9 downregulation and increases cell apoptosis [63].

Hypoxia-reoxygenation was shown to reduce miR-24 expression in rat cardiomyocytes, whereas miR-24 mimics increase Bcl-2 protein levels and decrease apoptosis [64]; equally important, Mitogen-Activated Protein Kinase 14 (MAPK14) is a target gene of miR-24 and its expression is negatively regulated by this miR [64]. Notably, miR-135a overexpression is known to decrease cell apoptosis, lactate dehydrogenase levels, Troponin I, and inflammation following isoproterenol treatment [65]; luciferase assay analyses validated Toll-like receptor 4 (TLR4) as the specific target gene of miR-135a [65]. RT-qPCR and immunoblot analyses revealed that oxygen-glucose deprivation/reperfusion (OGD/R) injury significantly enhances miR-210 in primary cardiomyocytes. Decreased caspase-3 activity and cell apoptosis was detected in cells transfected with a miR-210 mimic; furthermore, the transcription factor E2F3, known to trigger cell apoptosis [66], is one of target genes of miR-210 [67].

Emerging evidence indicates a significant downregulation of miR-26a-5p expression in mice undergoing ischemia/reperfusion injury, as well as in cardiomyocytes undergoing hypoxia-reoxygenation. Overexpression of miR-26a-5p significantly inhibits cardiomyocyte apoptosis and improves cardiac function by repressing Phosphatase and tensin homolog (PTEN), thereby activating the PI3K/AKT signaling pathway; thus, miR-26a-5p protects cardiac function via regulating PTEN/PI3K/AKT upon ischemia/reperfusion injury [68]. Wang et al. demonstrated that miR-369 overexpression reduces cell apoptosis, inflammation, and is accompanied by decreased caspase-3 activity, secretion of interleukin (IL)-6, IL-1β, and TNF-α by suppressing Transient Receptor Potential Cation Channel Subfamily V Member 3 (TRPV3) [69]. When overexpressed in rat cardiomyocytes under hypoxia-reoxygenation conditions, miR-129-5p decreases cell death by targeting HMGB1 [70]. miR-147 was shown to be downregulated after hypoxia in rat cardiomyocytes and, in vivo, in a rat model of myocardial infarction; overexpression of this miRNA preserves cardiac function by silencing homeodomain interacting protein kinase 2 (HIPK2) [71]. miR-184 depletion in vitro was found to decrease cleaved caspase-3 and Bax and attenuate cell death in cardiomyocytes by targeting F-box protein 28 (FBXO28) under hypoxic conditions [72]. Liu et al. observed that the overexpression of miR-223 reduces cell apoptosis by inhibition of poly (ADP-ribose) polymerase 1 (PARP-1) in rats with myocardial infarction and in hypoxia-treated neonatal rat cardiomyocytes (NRCMs); PARP-1 is a downstream target of miR-223 and these researchers found that silencing PARP-1 can protect cardiomyocytes from hypoxia [73].

There is an interaction between the lncRNA taurine up-regulated gene 1 (TUG1) and miR-142-3p [74]. High mobility group box 1 protein (HMGB1) and Ras-related C3 botulinum toxin substrate 1 (Rac1) are key regulators of apoptosis. Overexpression of miR-142-3p can preserve cardiomyocytes under hypoxia from apoptosis by inhibiting HMGB1 and Rac1 [75].

Dual-luciferase reporter assay revealed that the secreted protein acidic and rich in cysteine (SPARC) is a direct target of miR-29b-3p [76]. Hypoxia decreases miR-29b-3p expression in H9c2 cardiomyoblasts and increases cell apoptosis. Overexpression of miR-29b-3p can protect H9c2 cells from apoptosis by reducing the TGF-β1/SMAD pathway and SPARC [76].

## 4. miRNAs and Cardiomyocyte Autophagy

Autophagy (from the Ancient Greek αὐτόφαγος autóphagos, meaning “self-devouring”) is a highly conserved process in which cells eliminate cytoplasmic components, such as damaged proteins, substrates, and organelles in a lysosomal-dependent-way [77,78,79,80]. Autophagy is a pro-survival mechanism, activated through two major pathways including mTOR (mammalian or mechanistic target of rapamycin) and Bcl2 [81,82], that plays important roles in maintaining homeostasis [83]. Hypoxia, nutrient starvation, toxic agents, and stress can induce autophagy [84,85], a process that is known to have beneficial effects on myocardial cells, especially during the reperfusion stage [86,87,88,89].

There are numerus miRNAs that protect myocardial injury by regulating autophagy (Table 1). For instance, miR-34a attenuates autophagy by decreasing the expression of Lc3-II, p62, and TNF-α in neonatal rat cardiomyocytes in hypoxia-reoxygenation conditions [90].

In anoxia/reoxygenation (A/R)-treated rat cardiomyocytes and infarcted murine hearts, inhibition of miR-429 by antagomiR-429, increases the expression of MO25, LKB1, pAMPKα, ATG13, p62, and LC3B-I/II. Overexpression of miR-429 induces cell apoptosis and reduces autophagy. Antagonism of miR-429 improves hypoxia injury by enrichment of MO25/LKB1/AMPK mediated autophagy [91].

Autophagy-related 7 (ATG7) was identified as a target gene of miR-542-5p in H9c2 cardiomyoblasts [92], whereas the overexpression of miR-208a-3p indirectly upregulates autophagy related 5 (ATG5) [93] by downregulating Programmed Cell Death 4 (PDCD4) [94]. In a rabbit model of myocardial infarction and in H9c2 rat cardiomyoblasts, miR-145 was shown to have a cardioprotective effect through the induction of cardiomyocyte autophagy by targeting fibroblast growth factor receptor substrate 2 (FRS2). Downregulation of miR-142-5p and miR-126 stimulate autophagy by increasing the levels of beclin-1 [95,96], a major player in cardiac autophagy [97]. Overexpression of miR-204 targets beclin-1 and blocks the transformation of LC3-I to LC3-II; miR-204 attenuates apoptosis via targeting SIRT1 in H9c2 cells under hypoxia-reoxygenation conditions [98,99].

Heat Shock Protein Family A member 5 (HSPA5) and MAPK-mTOR signaling are downstream of miR-199a: the downregulation of miR-199a suppresses autophagy by inhibiting HSPA5 and MAPK-mTOR signaling in rat cardiomyocytes under starvation [100]. The miR-212/miR-132 family directly targets the pro-autophagic FoxO3 transcription factor and overexpression of these miRNAs leads to an impaired autophagic response [101]; similarly, Lv et al. observed upregulation of miR-302a-3p in mice undergoing ischemia/reperfusion, and miR-302a-3p upregulation inhibits FOXO3 [102]. Intriguingly, miR-19a decreases cell apoptosis and necrosis via repression of Bim, a proapoptotic protein, and switches on autophagy in rat cardiomyocytes under hypoxia [103]; equally important, miR-30e-3p promotes cardiomyocyte autophagy and inhibits apoptosis by indirectly regulating the expression of Egr-1 (Early growth response-1), a zinc finger transcriptional protein that has been associated with cardiovascular disorders [104], in an ischemic/hypoxic environment [105]. Most recently, the inhibition of miR-17-5p was shown to inhibit myocardial autophagy through targeting STAT3 [106].

Long non-coding RNAs (lncRNAs) have been extensively investigated in cardiomyocyte autophagy: the lncRNA DCM-related factor (DCRF) regulates cardiomyocyte autophagy by targeting miR-551b-5p [107], the lncRNA autophagy promoting factor (APF) regulates autophagy and myocardial infarction by targeting miR-188-3p [108], and the lncRNA 2810403D21Rik/Mirf (myocardial infarction-regulatory factor) acts as a competitive endogenous RNA (ceRNA) of miR-26a: downregulation of 2810403D21Rik/Mirf results in upregulation of miR-26a to promote autophagy by targeting ubiquitin specific peptidase 15 (Usp15) [109].

The main miRs involved in the regulation of cardiomyocyte apoptosis, necrosis, and autophagy are reported in Table 1.

**Table 1 cells-11-00983-t001:** Main miRNAs linked to regulation of apoptosis, necrosis, and autophagy in cardiovascular disorders.

Mechanisms	miRNA	Target Gene(s)	References
**Apoptosis**	miR-1	Bcl2	[57]
	miR-7a-5p	VDAC1	[61]
	miR-9	CDK8	[63]
	miR-24	Mapk14	[64]
	miR-26a-5p	PTEN/PI3K/AKT	[68]
	miR-29b-3p	SPARC/TGF-β1/SMAD	[76]
	miR-122	GATA-4	[58]
	miR-129-5p	HMGB1	[70]
	miR-133a	TAGLN2, HSP60, HSP70, Apaf1, Caspase3,8,9	[53,54,55,56]
	miR-135a	TLR4	[65]
	miR-137-3p	KLF15	[60]
	miR-142-3p	HMGB1, Rac1	[75]
	miR-147	HIPK2	[71]
	miR-184	FBXO28	[72]
	miR-208a	APC	[59]
	miR-210	E2F3	[67]
	miR-223	PARP-1	[73]
	miR-369	TRPV3	[69]
	miR-613	PDCD10	[62]
**Necrosis**	miR-30b	CypD	[18]
	miR-103/107	FADD	[15]
	miR-155	RIP1	[23]
	miR-325-3p	PIPK3	[22]
	miR-874	Caspase-8	[20]
	miR-873	P53	[16]
	miR-2861	ANT1	[26]
**Autophagy**	miR-17-5p	STAT3	[106]
	miR-19a	Bim	[103]
	miR-26a	Usp15	[109]
	miR-30e-3p	Egr-1	[104]
	miR-34a	Lc3-II, p62, TNF-α	[90]
	miR-126	Beclin-1	[96]
	miR-132	FoxO3	[101]
	miR-142-5p	Beclin-1	[95]
	miR-145	FRS2	[110]
	miR-199a	HSPA5/MAPK-mTOR	[100]
	miR-204	SIRT1	[98,99]
	miR-208a-3p	PDCD4	[94]
	miR-212	FoxO3	[101]
	miR-302a-3p	FoxO3	[102]
	miR-429	MO25/LKB1/AMPK	[91]
	miR-542-5p	ATG7	[92]
	miR-551b-5p	PCDH17	[107]

## 5. Clinical Studies: miRNAs as Biomarkers and Potential Therapies

Cardiac troponin I (cTnI), creatine kinase isoenzyme (CKMB), and brain natriuretic peptide (BNP) are extensively used for diagnosis of ischemic cardiovascular disorders [111]. Recently, miRNAs have been identified as reliable biomarkers for the early detection of cardiovascular diseases. For instance, several studies have shown that the expression of miR-499, miR-636, miR-380, miR-133a, miR-17, miR-21, miR-29b, miR-192, miR-194, miR-499, miR-1915, miR-34a, miR-423, miR-328, miR-134, miR-1254, miR-1, miR-181c, miR-208b, miR-566, miR-7-1, miR-92a, miR-455-3p, miR-126, miR-423-5p, miR-636, miR-486, and miR-1291 is increased in patients with acute myocardial infarction [112,113,114,115,116,117,118,119,120,121]. Equally important, plasma levels of miR-18a, miR-26b, miR-106a, miR-30e, miR-27a, and miR-199a have been found to be downregulated and miR-30d, miR-126, miR-1254, miR-37, miR-30c, miR-223-3p, miR-301a-3p, miR-210, miR-145-5p, miR-29a-3p, miR-1306-5p, miR-26b-5p, miR-199a-3p, miR-92a-3p, miR-146a, and miR-221 are upregulated in patients with heart failure (Table 2) [122,123,124,125,126,127,128,129,130,131,132,133].

The US Food and Drug Administration has already approved various RNA-based drugs targeting cardiovascular diseases and the pharmaceutical development of miRNA therapeutics is also in progress. In order to advance towards actual clinical application, proof of concept and safety evaluations must be carried out in large animal models, such as pigs and nonhuman primates. To study myocardial infarction, pigs are ideal for mimicking human heart disease as the porcine heart resembles the human heart in terms of weight, heart rate, and blood pressure [134].

Ex vivo models, including engineered heart tissue (EHTs) and living myocardial slices derived from human cells or tissues, are used as a bridge between in vitro and in vivo studies. Differentiated cardiomyocytes from human induced pluripotent stem cells (hiPSC) do not fully recapitulate the complex intercellular interactions observed in the whole human heart. Instead, several studies have shown that EHTs can recapitulate chronic heart disease phenotypes and miRNA-based drug development [135,136,137]. 

In an in vivo study carried out in mice, miR-92a was found to be overexpressed following cardiac ischemic injury; strikingly, administering antimiR-92a encapsulated in bioabsorbable and biocompatible microspheres via intracoronary injections in a swine model of myocardial infarction substantially improved angiogenesis [138]. A miR-92a inhibitor (Drug: MRG-110) intending to promote angiogenesis is currently under investigation in phase I clinical trials (Clinical Trial Identifiers: NCT03603431 and NCT03494712).

Another miRNA that is undergoing a Phase Ib clinical trial is miR-132 (Clinical Trial Identifiers: NCT04045405). Foinquinos et al. reported strong evidence for therapeutic efficacy of a locked nucleic acid based antisense inhibitor of miR-132 (antimiR-132) in a swine model of heart failure [101,139]. A miR-132 inhibitor (known as CDR132L) significantly preserves cardiac function and reverses cardiac remodeling in heart failure patients [140].

## 6. Conclusions

The overview on miRs modulating cardiomyocyte death presented here underlies the active research in this area and embodies a useful guide for the investigators in this field. Considering that targeting miRs and other non-coding RNAs represents a specific strategy to counteract cardiomyocyte death, we anticipate substantial research in this direction in the next years.

## Figures and Tables

**Figure 1 cells-11-00983-f001:**
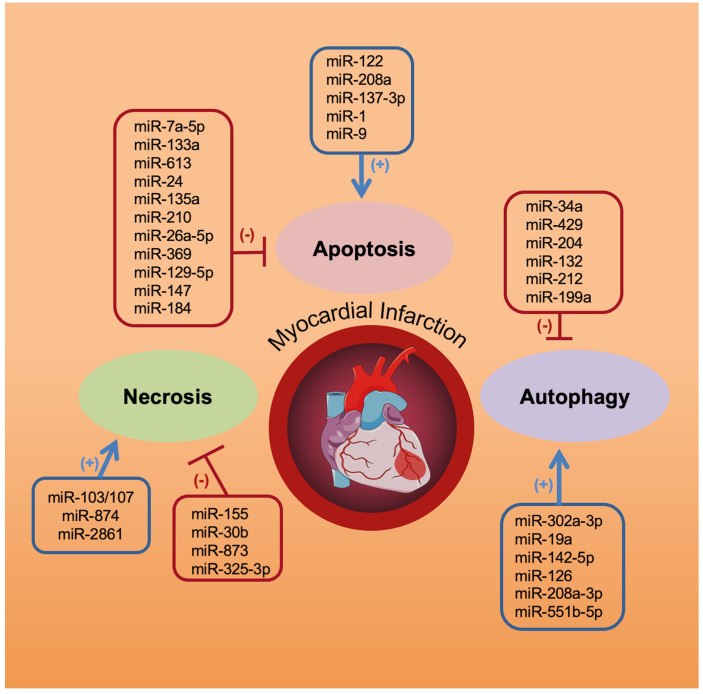
Schematic diagram depicting miRNAs orchestrating myocardial cell apoptosis, necrosis, and autophagy in myocardial infarction.

**Table 2 cells-11-00983-t002:** Diagnostic relevance of miRNA in acute myocardial infarction and heart failure.

Disease	miRNA	Expression	References
**Myocardial Infarction**	miR-1	Upregulation	[115,118]
	miR-21	Upregulation	[121]
	miR-29b	Upregulation	[115]
	miR-133	Upregulation	[113,114]
	miR-208b	Upregulation	[119]
	miR-221-3p	Upregulation	[120]
	miR-328	Upregulation	[117]
	miR-423	Upregulation	[116]
	miR-499	Upregulation	[112,113]
**Heart Failure**	miR-18a	Upregulation	[122]
	miR-21	Upregulation	[123]
	miR-26b	Upregulation	[124]
	miR-30c	Upregulation	[125]
	miR-30d	Downregulation	[126]
	miR-126	Downregulation	[127]
	miR-182	Upregulation	[128]
	miR-210	Upregulation	[129]
	miR-223	Downregulation	[122]
	miR-423	Downregulation	[130]
	miR-499	Upregulation	[131]
	miR-652	Downregulation	[122]
	miR-1254	Upregulation	[132]
	miR-1306	Downregulation	[133]

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
