# Peer review of "Functional Role of microRNAs in Regulating Cardiomyocyte Death"

_cells, 2022, doi:10.3390/cells11060983_

Round 1
Reviewer 1 Report
This paper lacks new ideas. There have been many reviews in this direction. Compared with the published review papers, there is no new summary of this field.
In this review, the authors described representative findings in miRNA-mediated cardiomyocyte death. The regulatory roles of miRNA in cardiac fibroblasts and endothelial cells were also discussed. However, there have been numerous reviews with similar topic. This review just simply listed the published results. It does not show enough novelty. There are some concerns as described below.
1. It is better to generalize and classify the miRNA-regulated signal pathways. Cross-talk between miRNAs and ceRNAs, extracellular miRNAs as well as clinical application of miRNAs should also be involved in this review.
2. This paper mainly focuses on the regulation mechanism of cardiomyocytes death. The author also describes fibroblasts and endothelial cells, which is not consistent with the theme. It is suggested to remove this part.
3. The English grammar could be improved throughout the manuscript.
Author Response
We thank this Reviewer for the helpful comments/suggestions. As recommended by the handling Academic Editor, we have removed the sections on fibroblasts and endothelial cells and we have added a paragraph discussing potential therapeutic applications of miRNA-based approaches; the Editor also stated that “As a review article, the expectation is not necessarily to provide "new ideas" or "novelty" per se, and it is perfectly acceptable to synthesize a great deal of published work and provide clarity to a complex topic.”
Reviewer 2 Report
The manuscipt by Kansakar et al is a comprhensive review on functional involvement of miRNAs in the regulation of cardiomyocyte death, a central event in acute myocardial infarction, ischemia/reperfusion, and heart failure.
Although the manuscript is well presented and written, it does not go beyond the simple literature report of the miRs that are linked to cardiomyocyte death.
However, the manuscript could be much improved if the authors discuss the role of those miRs either as specific markers or indicators of repsone to therapy. A better conclusion paragraph is also required to summarize the reported information.
Author Response
Thank you very much for your insightful suggestion; we have added a paragraph discussing potential therapeutic applications of miRNA-based approaches; and we have added a conclusion paragraph, as well.
Round 2
Reviewer 1 Report
The manuscript has been imporved, I have no further comments.